# Abortion Incidence among Young Women in Urban Slums and Non-Slums in Kinshasa, DR Congo

**DOI:** 10.3390/ijerph21081021

**Published:** 2024-08-02

**Authors:** Glory B. Nkombondo, Francis K. Kabasubabo, Pierre Z. Akilimali

**Affiliations:** 1Kinshasa School of Public Health, University of Kinshasa, Kinshasa P.O. Box 11850, Democratic Republic of the Congo; glorynk2005@gmail.com; 2Patrick Kayembe Research Center, Kinshasa School of Public Health, University of Kinshasa, Kinshasa P.O. Box 11850, Democratic Republic of the Congo; fkabasu13@gmail.com; 3Department of Nutrition, Kinshasa School of Public Health, University of Kinshasa, Kinshasa P.O. Box 11850, Democratic Republic of the Congo

**Keywords:** incidence of abortion, slum, Kinshasa, closest confidants

## Abstract

Background: Worldwide, around 73 million induced abortions take place every year. Of these, 45% are unsafe and can lead to complications. The evolution of the legal and practical landscape of abortion in the Democratic Republic of the Congo (DRC) over the last few years necessitates a re-examination of the experience of induced abortion, leading this study to measure the incidence of abortion among young women (15 to 29 years of age), as well as the heterogeneity of this problem according to the residence of these young women (slum vs. non-slum areas). Methodology: We used representative survey data on women aged 15–49 in Kinshasa, collected from December 2021 to April 2022. The survey included questions about the respondents’ and their closest confidants’ experience of induced abortion, including the methods and sources used. We estimated abortion incidence and heterogeneity over one year based on residence in the city of Kinshasa according to sociodemographic characteristics. Results: The fully adjusted one-year friend abortion rate in 2021 was 131.5 per 1000 (95% CI: IQR 99.4–163.6). These rates were significantly higher than the corresponding estimates of respondents. The incidence of induced abortion for respondents was 24.4 per 1000 (95% CI: 15.8–32.9) abortions per 1000 women. The incidence rates of induced abortion were much higher among the respondents residing in slums than among those residing in non-slums (29.2 vs. 13.0 per 1000; *p* < 0.001). Slum respondents indicated higher use of non-recommended methods than non-slum respondents. Conclusions: More precise estimates of the incidence of abortion indicate that the incidence rate of abortion was higher among young women residing in slums who were unmarried and had no children. These incidences were higher among confidants than among respondents. There is still a lot of work to be done to fulfill the obligations outlined in the Maputo Protocol. The aim is to decrease the occurrence of unsafe abortions and their associated effects.

## 1. Background

Induced abortion is a prevalent occurrence in global reproductive health, with an annual rate of around 39 abortions per 1000 women aged 15–49 years [1]. When conducted in accordance with medical protocols and under the supervision of healthcare practitioners, induced abortion appears to be highly secure. Nevertheless, unsafe abortion is responsible for approximately 8% of maternal fatalities worldwide [2].

The majority of these fatalities transpire in the Global South, particularly in sub-Saharan Africa, where a substantial number of abortions transpire outside the established healthcare system. The reasons for this are legislative constraints, social disapproval, financial burden, and the scarcity or unavailability of secure abortion facilities [3]. An analysis of abortion rates and safety trends is necessary to track advancements and provide information for ongoing reforms due to the changing legal and practical situation surrounding abortion in the Democratic Republic of the Congo (DRC) in recent years. The availability of safe abortion in the DRC has been limited in the past due to various factors [4]. These include conflicting laws [5] that have caused uncertainty about its legal status, a shortage of trained medical professionals and equipment to perform the procedure [6], inadequate awareness among potential clients about where to access abortion services [7], and societal stigma surrounding the procedure [8]. The ratification of the Maputo Protocol in 2018 and its publishing in the National Gazette should have resulted in the decriminalization of abortion and the definitive resolution of its legal status. According to Article 14 (2) (c) of the Maputo Protocol, States Parties are required to implement necessary measures to safeguard women’s reproductive rights. This includes allowing medical abortion in situations involving sexual assault, rape, incest, and when the mother’s mental and physical health, as well as the life of the mother or the fetus, are at risk due to the continuation of the pregnancy [9]. 

Unsafe abortion contributes significantly to maternal morbidity and mortality, accounting for approximately 10% of maternal deaths in Africa [2] and especially among individuals residing in socioeconomically deprived or impoverished communities. A significant majority of the impoverished urban population in numerous sub-Saharan nations reside in slums, which are frequently isolated from official public services, including FP programs [10]. Municipal authorities frequently fail to recognize slums as essential components of urban areas. Consequently, individuals living in slums are often excluded from official systems that provide services [10]. Due to the predominantly governmental nature of most FP programs, the urban underprivileged, especially those residing in slums, frequently face barriers while attempting to receive these services. Considering that the majority of urban population growth in the future will occur in developing countries, it is crucial to conduct research on the disparities in family planning outcomes between slum and non-slum residents. This research should also investigate the role of access to family planning services in contributing to these disparities, given the significant health implications of urbanization.

The growth of the urban population imposes an extra strain on the already limited resources of local governments [11]. Insufficient knowledge exists regarding the diversity of reproductive health outcomes, including fertility desires, family planning utilization, and unintended pregnancies, in both urban slums and non-slum areas, despite the associated health consequences. Limited information exists regarding the induced abortion and availability of reproductive health services for individuals living in urban poverty. Limited research has been conducted on the utilization of contraception among women residing in African slums and non-slum areas [12,13,14]. Akilimali published an abortion incidence rate in Kinshasa of 105.3 per 1000 women [15]. These alarming statistics have prompted unprecedented policy action to reduce the morbidity and mortality associated with unsafe abortion in the DRC. However, the incidence rate was not presented in relation to the two groups of neighborhoods (slum vs non-slum), despite the existence of compelling grounds for the variability between these two types of populations in the city of Kinshasa.

The aim of this research was to examine the heterogeneity of abortion incidence between slum and non-slum residents, specifically among women aged under 30 years. This age group is currently undergoing the emergence of sexual urges and the chance to explore their sexuality [16], yet emotional maturity is typically not yet fully established. Consequently, teenagers and young individuals are more prone to participate in sexual activities that can be classified as “high-risk”. These actions refer to engaging in sexual intercourse without using a condom, whether it be vaginal, anal, or oral sex, or having sex while under the influence of drugs or alcohol. These behaviors can result in unplanned pregnancy and/or the spread of sexually transmitted illnesses [17]. The heightened probability of participating in high-risk sexual behavior renders sexual health a crucial health concern in those under the age of 30.

## 2. Methodology

### 2.1. Sampling and Data Collection

Data were used from the Performance Monitoring for Action (PMA) study, which was conducted in Kinshasa and Central Congo between December 2021 and April 2022. PMA conducts nationally or regionally representative reproductive health surveys in nine countries in Africa and Asia [18]. The PMA DRC carried out surveys in two provinces, Kinshasa and Kongo Central, utilizing a two-stage cluster sampling methodology. The selection of clusters was conducted using probability proportional to size sampling. Within each cluster, a complete enumeration of all households was conducted, and a random selection of 35 households was made to obtain a representative sample. All females between the ages of 15 and 49 who were listed in the household survey were asked to take part in the women’s survey throughout the 2019–2020 period (Phase 1). The ladies were tracked on a yearly basis during Phase 2 (2020–2021) and Phase 3 (2021–2022). Out of the initial database of 2326 women in Kinshasa, we specifically chose 1397 women who were between the ages of 15 and 29. For more details on the PMA selection procedures, see www.pmadata.org/data/survey-methodology, accessed on 5 April 2024. Ethical approval for this study was obtained from the Institutional Review Board at the Johns Hopkins University Bloomberg School of Public Health (approval #14590) in November 2021 and from the Comité d’Éthique at the Kinshasa School of Public Health (approval #ESP/CE/159B/2021) in October 2021. More details on sampling and data collection are described in our previous article [15].

A slum household is defined as one that is situated in a city and lacks at least three of the following amenities: (1) access to a reliable water source (such as a piped connection to the house or plot, a public tap or standpipe, a tube well or borehole, or a protected dug well); (2) access to improved sanitation facilities (such as a flush or pour flush system connected to a piped sewer, a septic tank or pit latrine, or a ventilated improved pit latrine); (3) sufficient living space (with no more than three individuals per room); and (4) housing durability (i.e., the condition of the floors, walls, or roof of the household being natural, basic, or completed). UN-Habitat suggests that enumeration areas should be utilized for identifying slum neighborhoods as they serve as the smallest household clusters in numerous countries and exhibit a considerable degree of homogeneity [19]. The slum families were grouped together at the enumeration area or cluster level to create the slum neighborhood variable that was utilized in this study. According to UN-Habitat [19], a cluster was classified as a slum neighborhood if it contained 50% or more households that were considered slum households.

### 2.2. Measures

The abortion experience of both the friends and respondents was evaluated by two sets of inquiries: one inquiring about any previous actions taken to terminate a pregnancy, and another inquiring about any previous actions taken to manage a delayed menstrual cycle. To ascertain the reason for the action taken, a subsequent inquiry was made regarding whether it was motivated by concerns of potential pregnancy during menstrual regulation. This study defines abortion as any deliberate action to terminate a pregnancy, whether it is through successful methods of termination or by successful methods of inducing menstruation due to concerns about pregnancy. The following inquiries examined the timing of the most recent abortion encounter, the techniques and origins employed, and whether any complications arose.

We utilized data pertaining to reported sources and techniques of abortion in order to construct a categorical metric that determines whether the abortion procedure employed a recommended method and/or source. This measure was derived from the latest indicator of abortion safety provided by the World Health Organization (WHO) in 2022. According to the World Health Organization (WHO), safe abortions are defined as procedures conducted by skilled practitioners using approved techniques, such as surgical abortion or the use of misoprostol alone or in combination with mifepristone. Safe abortions can be performed by trained providers using suggested procedures. However, the least safe abortions are carried out by untrained providers using non-recommended methods.

Using our data, which did not account for provider training, we categorized abortions obtained from various public or private clinical sources, including national hospitals, regional hospitals, government health centers, family planning clinics, maternity clinics, community health workers, private hospitals, NGOs, private health centers, private practices, private physicians, mobile nurses, and community health workers, as being sourced from a recommended provider. All sources, except for those providing information on medical abortion pills, were deemed non-recommended. In order to incorporate the World Health Organization’s most recent abortion recommendations, which state that self-managed medical abortion can be considered safe in some circumstances, we classified all abortions involving medical abortion pills (such as misoprostol with or without mifepristone) from any source as being associated with a recommended source [20]. Despite our limited knowledge on the precise details of the surgical or medicinal abortion pill regimen, we categorized both surgical abortions and medical abortion pills as approved procedures. Any techniques other than pills, injections, or traditional procedures were categorized as not recommended. As a result, our categories evaluated whether the abortion utilized a “recommended method and source”, a “non-recommended method or source”, or a “non-recommended method and source”.

This study incorporated various sociodemographic and reproductive health factors for both the respondents and their friends. These factors included age, education level, marital status, province of residence (assuming that the friend lived in the same province as the respondent), parity, current use of contraceptives, and current use of long-acting reversible contraceptives (LARCs), such as intrauterine devices (IUDs) and implants. We conducted a focused analysis on the frequency of LARCs (long-acting reversible contraceptives) usage. This was initiated because individuals and their acquaintances are likely to provide more precise information regarding these methods, which are used over a longer period of time, compared to methods that include behavior, sexual activity, or user control, which may be initiated and discontinued more often. Prior studies have indicated that by comparing the prevalence of long-acting reversible contraception (LARC) among survey participants and their friends, we can verify the premise that surrogate samples in social network-based approaches are similar and that less sensitive reproductive health behaviors are shared [21]. We also analyzed the wealth tertiles of respondents by using a principal components analysis. This analysis was based on information about household assets, water, sanitation, and construction materials. The methodology we used was similar to that used in the Demographic and Health Surveys. Unfortunately, we were unable to acquire the household information for the friends, which prevented us from analyzing the correlation between wealth and abortion trends in the proxy sample.

### 2.3. Analysis

To examine and adjust for potential violations of assumptions (selection and transmission biases) in the friend data, we followed a five-step process described in our previous article [15].

We compared the fully adjusted friend estimates to the respondent rates by assessing whether the confidence intervals overlapped, as no appropriate statistical test was available due to the post hoc nature of the transmission bias adjustment described above.

We then analyzed the distribution of abortion methods and sources for both the respondents and their friends. In addition, we examined the sociodemographic characteristics associated with the use of non-recommended abortion methods and sources, which may pose the greatest risk of adverse outcomes for both the respondents and their friends. Design-based F-tests were used to assess the statistical significance of these differences.

All analyses were conducted with Stata version 17 (Stata Corp, College Station, TX, USA). We constructed survey design weights separately for each province to account for the complex sampling strategy, reflecting the inverse probability of selection and nonresponse. We computed robust standard errors to adjust for clustering. The design weights were then combined with the post-stratification weights for the proxy sample.

## 3. Results

### 3.1. Sociodemographic Characteristics of Respondents and Confidants

Table 1 shows the distribution of respondents by sociodemographic characteristics and type of residence. Among 1397 women, 74.5% were living in slum neighborhoods. Slum women and non-slum women were similar in terms of age, Recent Contraceptive Use, Use of Modern Contraceptive Methods, and reporting a close female friend. However, regarding education, more women in non-slum neighborhoods than those in slum neighborhoods had attended higher levels of schooling (28% vs. 19%). A higher percentage of women living in non-slum neighborhoods were nulliparous women than of those living in slum neighborhoods (75% vs. 62%; *p* < 0.001). However, there was a lower percentage of women in unions in non-slum neighborhoods than of those in slum neighborhoods (24% vs. 13%). Other characteristics are described in Table 1.

Our sample included 1397 women with a mean age of 21.9 years. Of these, 21% were married, 36% had children, and 97% had attained a higher level of education. At the time of the survey, 42% of the women were using contraception, and 8% were using long-acting reversible contraceptives (LARCs). In addition, 70% of the respondents lived in slum areas. See Table 2 and Table 3 for more details.

Overall, about 64% (890/1397) of the respondents reported having a confidant living in the DRC. The confidants (according to the unadjusted and adjusted proxy samples) were very similar to the respondents. The adjusted sample of confidants was slightly younger than the sample of respondents (Table 2). Having at least one child, educational attainment, current contraceptive use, and long-acting contraceptive use were similar between the respondents with 0 confidant and those with 1 + confidants (Table 3). After adjustment, it was found that the confidants were slightly more likely to be married than the respondents.

### 3.2. Sociodemographic and Economic Characteristics of Respondents Who Reported an Abortion and Shared It with a Confidant, According to Type of Abortion

The slum-dwelling respondents were more likely to have shared their end-of-pregnancy experiences with their confidants than the non-slum-dwelling respondents (59% vs. 50%; *p* = 0.005). Overall, when end-of-pregnancy experiences and menstrual regulation were combined, it was found that the slum-dwelling respondents shared more with their confidants than the non-slum-dwelling respondents (57% vs. 42%; *p* < 0.001).

To account for transmission bias, we examined the percentage of respondents who shared their abortion experience with their closest friend among those who reported having had an abortion and a close friend. Among this group, 56% reported sharing their abortion experience (ending pregnancy) with their closest friend, while 59% reported sharing their menstrual regulation (period regulation) experience with their closest friend (Table 4).

### 3.3. Incidence of Induced Abortion among Respondents Aged 15–29 and Their Confidants

The incidence rates of induced abortion were significantly higher among confidants than among respondents. The one-year incidence of induced abortion among respondents in 2021 was 24.4 per 1000 women (95% CI: 15.8–32.9), while the fully adjusted abortion rate among confidants was 131.5 per 1000 women (95% CI: 99.4–163.6). The highest abortion rates among both respondents and confidants were observed among women aged 20–29 years and single women.

Abortion rates were highest among respondents with higher levels of education; however, the abortion rate among confidants was lowest among those most highly educated. In addition, the incidence rate of induced abortion was significantly higher among respondents living in slums than among those living in non-slum areas (29.2 vs. 13.0 per 1000; *p* < 0.001) (Table 5).

### 3.4. Characteristics of Abortions Achieved by Respondents and Their Confidants

Among the respondents in Kinshasa, surgery was the most common method of abortion (35%), followed by medical abortion pills (28%). Injections (23%) and other non-medical abortion pills (15%) were also commonly used. The most common source of care was private facilities (45%), followed by pharmacies (36%). The relative frequency with which women relied on specific methods and sources was similar between provinces, but they involved more non-recommended methods (Table 6). 

### 3.5. Characteristics Related to the Safety of the Last Abortion among Respondents and Their Confidants

Abortions using recommended methods and sources accounted for 57% of the respondents’ abortions, with confidential data suggesting a similar level at 61%, which was not significantly different from respondents’ estimates. Approximately one in five abortions (20%) among respondents involved non-recommended methods and sources, with a corresponding estimate of 18% among the confidants (Table 7). Slum respondents indicated higher use of non-recommended methods than non-slum respondents (Figure 1).

## 4. Discussion

This study was conducted to describe and measure the incidence of abortion among young women (15 to 29 years of age), as well as the heterogeneity of this problem according to the residence of respondents (slum vs. non-slum areas). We performed a secondary analysis of data obtained from PMA surveys conducted in Kinshasa from December 2021 to April 2022. The analysis focused on abortions among women who are capable of carrying children. The objective was to ascertain the prevalence of abortions among this demographic and to examine the disparity in prevalence based on the young woman’s place of living (slum vs. non-slum areas). This study aimed to understand the experience of induced abortion among women residing in the slums of Kinshasa, where there is restricted availability of formal healthcare services.

Indeed, the prevalence of abortion was greater among individuals residing in slums (29.2 per 1000) than among those who did not reside in slums (13 per 1000), being more than double. These findings are consistent with the data presented in a study conducted by Bell SO et al. in Côte d’Ivoire in 2020. Their study found a rate of 29 per 1000 among women aged 15 to 49 [22]. The elevated prevalence of this phenomenon among young females residing in impoverished urban areas is believed to stem from the limited access to reproductive health services in disadvantaged regions, including family planning (FP) services, where abortion is the preferred method of contraception. Additionally, the restricted availability of FP services for young individuals further contributes to this issue. In 2015, a study conducted by Behera D et al. in Mumbai, India, also corroborated these findings. The study reported an incidence rate of 116 per 1000 [23], which was attributed to cultural factors that hinder the use of contraceptive methods and consequently contribute to a substantial number of unintended pregnancies. It is important to highlight that these abortions are frequently carried out by individuals living nearby, and they pose significant risks and hazards. Research has shown that slum populations experience higher levels of unmet need for family planning (FP) [24] services, making them more susceptible to resorting to induced abortion methods. In addition, reproductive services, such as abortion, are highly uncommon in slums [25,26].

The young women included in this investigation were mostly unmarried (more than three-quarters of our group). Among the confidants, unmarried women had a higher rate of resorting to abortions than married women (156 vs. 60). This occurrence can be attributed to the unfavorable societal view toward children born to unmarried parents [27]. These findings align with those of a study conducted by Santos et al. in Brazil in 2012, which revealed that unmarried women were approximately 3.6 times more likely to resort to unsafe abortion than married women [28]. The findings of Akilimali et al.’s studies on the entire sample align with our own results. In Kongo Central, the incidence rate among unmarried women was 29 per 1000, while among married women, it was 8 per 1000 [15].

Regarding economic status, those with a low economic level had a higher incidence rate of abortion (37 per 1000), which was four times more than those with a high economic level (9 per 1000). This disparity is attributed to the inability to meet the financial demands involved with caring for a newborn. This finding contradicts that of a previous study conducted in Abidjan, which found that individuals with a higher socioeconomic position had a greater occurrence rate than those from lower socioeconomic classes [29]. Poverty poses a barrier to the act of welcoming a new child [30].

It is crucial to highlight that 43% of these abortions were categorized as unsafe. Approximately 20% of abortions were performed using procedures and sources that are not recommended. The estimate for the confidants was 18%, which aligns with the findings of Bankole, A. et al.’s study conducted in sub-Saharan Africa in 2020. Specifically, that study revealed that three out of four (77%) abortions were unsafe [31]. In terms of the techniques employed, surgery was the most commonly utilized approach among the participants (35%), followed by the administration of medical abortion pills (28%). Private institutions were the predominant source of care, accounting for 45% of cases, followed by pharmacies at 36%. This is in contrast to the findings of Aké-Tano, P. et al.’s study conducted in Yamoussoukro in 2017, where the predominant strategy utilized by participants was the consumption of medical abortion pills, accounting for 92% of cases [32].

The findings of this study are unquestionably valuable and can be utilized nationwide, as well as in other countries impacted by abortions, particularly unsafe abortions, which are a major contributor to maternal mortality. The findings of this study will assist policymakers in implementing interventions aiming to improve access to reproductive health services for adolescent girls. The objective is to decrease the occurrence of unsafe clandestine abortions, which are a significant contributor to maternal morbidity and mortality in Kinshasa, particularly in disadvantaged neighborhoods. The potential for selection bias in our study is a concern, as we conducted a secondary analysis on an existing database and had no influence over the recruitment process for the subjects in our sample.

Due to social desirability bias, estimating abortion incidence rates among respondents would underestimate the abortion incidence. Using friends or confidants gives us the opportunity. Although we may have indirectly measured abortion through closest friends or confidants, we acknowledge the presence of bias. Nevertheless, the estimated value obtained through closest friends or confidants is more accurate than the estimate provided by the respondents. Another constraint arises from the inclusion of close friends/confidants who were not all under the age of 30, which complicates the comparison of incidence between the respondents and their friends. Furthermore, we lack data regarding the living arrangements of close acquaintances (whether they reside in slums or non-slum areas) in order to provide a comprehensive understanding of the occurrence of abortions among close friends or confidants. Nevertheless, we managed to mitigate this bias by accurately establishing inclusion criteria for picking our sample from the extensive PMA database.

## 5. Conclusions

This study found that the prevalence of abortion was greater among young women residing in impoverished urban areas, young women who were not married, those who had not yet become mothers, and those who did not utilize contemporary contraceptive techniques; these occurrences were marginally more frequent among acquaintances than among the survey participants. The incidence of induced abortion for respondents was higher in slum areas compared to non-slum areas. Slum respondents exhibited higher use of non-recommended methods than non-slum respondents.

There is still a significant amount of work that needs to be carried out to fulfill the obligations outlined in the Maputo Protocol. Specifically, efforts should be focused on establishing easily accessible services that offer complete abortion care with a strong emphasis on women’s needs. This is particularly important in slum areas, as it will help to decrease the occurrence of unsafe abortions and mitigate their negative effects.

## Figures and Tables

**Figure 1 ijerph-21-01021-f001:**
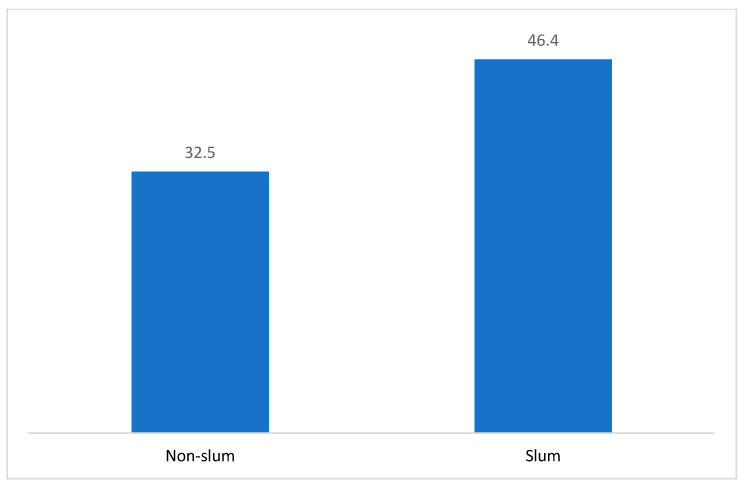
Percent of abortions involving non-recommended methods and sources among female respondents in Kinshasa, between slum and non-slum women.

**Table 1 ijerph-21-01021-t001:** Sociodemographic characteristics of women in Kinshasa by type of neighborhood, 2020 (weighted data).

	Non-Slum (n = 327)	Slum (n = 1070)	Overall (n = 1397)	*p*
	%	%	%	
Age Group (years)				0.886
15–19	37.6	36.8	37.0	
20–29	62.4	63.2	63.0	
Education				<0.001
None/primary	0.1	4.7	3.6	
Secondary	72.0	76.8	75.6	
Tertiary	27.9	18.5	20.9	
Marital Status				<0.001
Not married	87.4	75.7	78.7	
Married/in cohabitation	12.6	24.3	21.3	
Socioeconomic Status				<0.001
Low	8.9	39.4	31.6	
Middle	36.5	31.4	32.7	
High	54.6	29.2	35.7	
Parity				<0.001
0	74.8	61.8	65.1	
≥1 child	25.2	38.2	34.9	
Recent Contraceptive Use				0.197
No	63.4	58.0	59.4	
Yes	36.6	42.0	40.6	
Recent Use of LARCs				<0.001
No	97.7	92.1	93.6	
Yes	2.3	7.9	6.4	
Use of Modern Contraceptive Methods				0.183
No	77.4	72.7	73.9	
Yes	22.6	27.3	26.1	
Total	100.0	100.0	100.0	
Respondent reported having a close female friend				0.359
0 Confidant	38.9	31.7	33.6	
≥1 Confidant	61.1	68.3	66.4	
Total	100.0	100.0	100.0	

**Table 2 ijerph-21-01021-t002:** Distribution of characteristics of respondents aged 15–29 and their confidants aged 15–45 in Kinshasa, DRC *.

Characteristics	Respondents		Confidants Adjusted		Confidants Fully Adjusted **		*p*
	N	%	N	%	N	%	
Age Group (years)							
15–19	516	37.0	294	33.5	480	34.8	**<0.** **0001**
20–29	881	63.0	457	52.2	792	55.9	
30–39			100	11.5	100	7.5	
40–49			25	2.7	25	1.8	
Total	1397	100.0	876	100.0	1397	100.0	
Education							
None/primary	56	3.6	31	3.3	55	3.6	0.192
Secondary	1053	75.6	633	72.0	1030	74.3	
Tertiary	288	20.9	221	24.7	312	22.2	
Total	1397	100.0	885	100.0	1397	100.0	
Marital Status							
Not married	1088	78.7	661	75.1	1038	74.9	**0.051**
Married/in cohabitation	309	21.3	228	24.9	359	25.1	
Total	1397	100.0	889	100.0	1397	100.0	
Socioeconomic status							
Low	487	31.6	487	31.6			
Middle	464	32.7	464	32.7			
High	446	35.7	446	35.7			
Total	1397	100.0	1397	100.0			
Residence							
Non-slum	327	30.1	327	30.1			0.999
Slum	1070	69.9	1070	69.9			
Total	1397	100.0	1397	100.0			
Parity							
0	898	65.1	572	64.8	886	63.9	0.855
≥1 child	499	34.9	317	35.2	511	36.1	
Total	1397	100.0	889	100.0	1397	100.0	
Recent Contraceptive Use							
No	849	59.4	499	55.2	820	57.7	0.082
Yes	548	40.6	391	44.8	577	42.3	
Total	1397	100.0	890	100.0	1397	100.0	
Recent Use of LARCs							
No	1319	93.6	817	91.0	1292	92.0	**0.031**
Yes	78	6.4	73	9.0	105	8.0	
Total	1397	100.0	890	100.0	1397	100.0	

LARCs: long-acting reversible contraceptives. * Estimates weighted; Ns unweighted; bold indicates *p*-value for design-based F-test (reference respondents) less than 0.05. ** Estimates include respondent characteristics in place of “missing” confidants; post-stratification weights applied.

**Table 3 ijerph-21-01021-t003:** Distribution of the characteristics of female respondents aged 15 to 29 by the number of confidants in Kinshasa, DRC *.

	0 Confidant		≥1 Confidant		All Respondents		*p* Value
Characteristics	* N	%	N	%	N	%	
Age Group (years)							
15–19	183	37.6	333	36.7	516	37.0	0.623
20–29	324	62.4	557	63.3	881	63.0	
Education							
None/primary	21	3.2	35	3.6	56	3.6	0.177
Secondary	395	79.4	658	73.6	1053	75.6	
Tertiary	91	17.4	197	22.7	288	20.9	
Marital Status							
Not married	376	74.4	712	80.8	1088	78.7	0.011
Married/in cohabitation	131	25.6	178	19.2	309	21.3	
Socioeconomic status							
Low	180	34.1	307	30.3	487	31.6	0.055
Middle	184	35.0	280	31.5	464	32.7	
High	143	30.9	303	38.1	446	35.7	
Residence							
Non-slum	98	26.4	229	32.0	327	30.1	0.007
Slum	409	73.6	661	68.0	1070	69.9	
Parity							
0	313	61.9	585	66.7	898	65.1	0.127
≥1 child	194	38.1	304	33.3	498	34.9	
Recent Contraceptive Use							
No	321	62.8	528	57.6	849	59.4	0.142
Yes	186	37.2	362	42.4	548	40.6	
Recent Use of LARCs							
No	475	94.0	844	93.4	1319	93.6	0.371
Yes	32	6.0	46	6.6	78	6.4	
Use of Modern Contraceptive Methods							
No	383	75.2	654	73.2	1037	73.9	0.396
Yes	124	24.8	236	26.8	360	26.1	
History of Abortions							
No	414	97.9	829	97.3	1243	97.5	0.081
Yes	7	2.1	29	2.7	36	2.5	
Total	507	100.0	890	100.0	1397	100.0	

*: Ns were weighted, and %’s were unweighted. LARCs: long-acting reversible contraceptives.

**Table 4 ijerph-21-01021-t004:** Statistics among respondents who reported an abortion and having a confidant or best friend, percentage who shared it with their friend, overall and by abortion type.

		Ending Pregnancy		Period Regulation		Combined	
		%	N	%	N	%	N
Age							
	15–19	72.9	12	100.0	1	73.5	13
	20–29	53.3	80	57.9	10	49.3	90
Education							
	Never	100.0	1	69.2	8	100.0	9
	Primary	52.5	68	69.2	6	53.6	74
	Secondary/higher	65.5	23	46.5	5	49.1	28
Marital Status							
	Currently married/cohabiting	68.6	62	54.9	9	62.1	69
	Not married	32.8	30	66.3	5	34.2	34
Wealth Tertile							
	Poorest	55.9	39	18.7	5	50.8	44
	Middle wealth	55.9	29	18.7	5	50.8	34
	Wealthiest	51.1	24	73.0	4	54.9	28
Slum							
	Non-slum	49.8	22	0.0	4	42.1	26
	Slum	58.7	70	85.7	10	56.8	80
Total		56.3	92	59.0	14	52.6	106

**Table 5 ijerph-21-01021-t005:** Distribution of the incidence of induced abortion (per 1000) among respondents aged 15 to 29 and their confidants according to sociodemographic and economic characteristics in Kinshasa, DRC *.

		Respondent			Partially Adjusted Confidant **			Adjusted Confidant ***		
	Characteristic	Rate	95% CI		Rate	95% CI		Rate	95% CI	
Age Group (years)										
	15–19	21.1	10.9	40.8	84.8	48.1	121.4	139.9	79.4	200.3
	20–29	27.2	18.3	40.5	85.4	64.1	106.8	141.0	105.7	176.2
Education Level										
	None	16.9		49.5	129.0	27.9	230.2	212.9	46.0	379.8
	Primary	20.6	12.1	29.1	78.6	58.0	99.3	129.8	95.7	163.9
	Secondary/tertiary				74.8	33.1	116.5	123.4	54.6	192.2
Marital Status										
	Not married	24.6	13.5	35.7	94.2	69.8	118.7	155.5	115.1	195.8
	Married/in cohabitation	23.6	6.1	41.0	36.5	20.7	52.3	60.2	34.1	86.3
Socioeconomic Status										
	Low	36.6	14.7	58.5						
	Middle	29.2	13.8	44.7						
	High	9.4	1.2	17.5						
Residence										
	Non-slum	13.0	0.9	25.1						
	Slum	29.2	19.4	39.0						
Parity										
	0	24.1	11.6	36.6	76.6	50.0	103.2	126.5	82.6	170.4
	≥1 child	24.9	10.6	39.3	85.0	55.5	114.6	140.3	91.5	189.0
	Total	24.4	15.8	32.9	79.7	60.3	99.2	131.5	99.4	163.6

* Estimates weighted; Ns unweighted; bold indicates 95% confidence intervals do not overlap (reference respondents). ** Estimates include respondent characteristics in place of “missing” confidants; post-stratification weights and transmission bias adjustment applied. *** Applies transmission bias adjustment factor to partially adjusted rates.

**Table 6 ijerph-21-01021-t006:** Distribution of characteristics of recent abortions reported by respondents aged 15 to 29 and their confidants aged 15 to 49 in Kinshasa, DRC.

		Respondent		Friends Overall	
		%	N	%	N
All methods used (multiple select)					
	Surgery	34.9	56	43.9	100
	Mifepristone/misoprostol pills	28.0	51	25.3	56
	Other pills (identified)	15.3	33	16.4	52
	Unknown pill type	4.7	10	4.7	12
	Injection	23.2	46	14.0	38
	Traditional/other methods	8.5	15	15.1	33
	Do not know/No response	1.9	4	5.6	12
All sources used (multiple select)					
	Public facility	12.5	23	13.3	32
	Private facility	44.7	81	62.4	145
	Pharmacy	35.9	66	30.3	77
	Other non-clinical	10.4	20	12.0	26
	Do not know/No response	0.3	1	1.6	5

**Table 7 ijerph-21-01021-t007:** Distribution of the level of safety of the last abortion among respondents aged 15 to 29 and their confidants aged 15 to 45 in Kinshasa, DRC.

		Overall				Last 3 Years			
	Categories	Respondents		Confidants Adjusted **		Respondents		Confidants Adjusted *	
		%	N	%	N	%	N	%	N
Current WHO Safety Measures **									
	Safe (recommended methods and sources)	38.1	65	41.6	90	31.6	34	52.5	26
	Less safe (one criterion met)	41.4	77	40.3	92	44.1	49	34.9	17
	Unsafe (non-recommended methods and sources)	20.6	42	18.1	50	24.3	29	12.6	10
With New Self-Managed MA Reflected ***									
	Safe (recommended methods and sources)	57.2	99	60.7	132	51.8	55	64.5	31
	Less safe (one criterion met)	22.2	43	21.2	50	23.9	28	22.9	12
	Unsafe (non-recommended methods and sources)	20.6	42	18.1	50	24.3	29	12.6	10
Total		100	184	100	232	100	112	100	53

* Adjusted friend data include respondent abortion details for respondents who reported having no friends. ** Surgery and medication abortion from clinical source = safe. *** Surgery from facility and medication from any source = safe.

## Data Availability

The dataset used for analysis can be obtained upon reasonable request by writing an email to the corresponding author.

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
