# Peer review of "Abortion Incidence among Young Women in Urban Slums and Non-Slums in Kinshasa, DR Congo"

_ijerph, 2024, doi:10.3390/ijerph21081021_

Round 1

Reviewer 1 Report

Comments and Suggestions for Authors

Report of manuscript review

This study on Abortion Incidence among Young Women in Urban Slums and Non-Slums in Kinshasa, DR Congo is crucial to understand how women facing unintended pregnancy manage their situation. this in turn, helps to inform the decision make to take action in order to safe women life by providing safe induce abortion services. I would like to congratulate the authors for their effort. However, there are important aspects not considered in this study that could provide a more comprehensive understanding. While this study aimed to understand the experience of induced abortion among women residing in the slums of Kinshasa, where there is restricted availability of formal healthcare services, the author did not show, in the result, the experience of the participants based on categories slum and non-slum. In the table presented they explore types of induced abortion and show for whom they reported. What the importance of exploring to whom they reported? What benefit does this offer in terms of strategy of health improvement. These questions are important to be explained to make the reader in context. Bellow I present specific comments.

Abstract:

1.       normally, groups of age being used in sexual and reproductive health are 15-49 as women of reproductive age in general. When the study involves only young people the groups used are 15-19 for adolescents and 20-24 youth.

·       can the Autor explain the groups of age used in this study? What is the definition applied, from whom?

2.       The incidence rate of induced abortion among the confidants (close friends) in 2021 was 131.5 per 1000 (95% CI: IQR 99.4–163.6). These rates were significantly higher than the corresponding estimates of those surveyed.

·       Which estimates, which surveyed, and by whom?

3.       There is still a lot of work to be done to fulfill the obligations outlined in the Maputo Protocol. This includes establishing easily accessible services that offer complete abortion care with a focus on women's needs, particularly in slum areas.

·       from the result described before, it is not evident this statement the Autor made

Background

4.       An analysis of abortion rates and safety trends is necessary to track advancements and provide information for ongoing reforms due to the changing legal and practical situation surrounding abortion in the Democratic Republic of the Congo (DRC) in recent years.

·       can the author explain a bit more this story about abortion law before the changes and what changes are occurring now?

5.       A significant majority of the impoverished urban population in numerous sub-Saharan nations reside in slums, which are frequently isolated from official public services, including FP programs.

·       slums correspond to peripheral areas of the cities characterized by high population density which result in high demand in services which most the time do not meet integrally the need of that population. when the author says isolated from official services, is this similar to say that they walk long distances, for example 10 km, to find health facility? Is lack of health services infra-structure or the problem is the response to demand?

6.       Akilimali published abortion incidence in Kinshasa [9], These alarming statistics have prompted unprecedented policy action to reduce the morbidity and mortality associated with unsafe abortion in the DRC. Where can we see these alarming statistics in this manuscript? Can you include them to make more evident

7.       The aims of this research were to the heterogeneity of abortion incidence between slum and non-slum residence specifically among youth aged under 30 years.

Seems like something is missing in this passage

which concept of youth this study is using in this study, WHO or from other organization and why? see comment in abstract section

8.       These behaviors can result in unplanned pregnancy and/or the transmission of sexually transmitted illnesses. There is repetition of information

Methodology

9.       Data from the Performance Monitoring for Action (PMA) study were used

·       can the author explain more about this PMA

10.   Out of the initial database of 2326 women in Kinshasa, we specifically chose 1397 women who were between the ages of 15 and 29.

·       how these specific women were selected for this study? in the recommended place to see methodology selection process the explanation presented is about  selection process of all sample.

11.   This measure was derived from the latest indicator of abortion safety provided by the World Health Organization (WHO).

·       Indicate the year

12.   However, the least safe abortions are carried out by untrained clinicians using non-recommended methods.

·       What do we call clinicians

13.   Any techniques other than pills, injections, or traditional procedures were categorized as not recommended.

·       explain a bit more this statement to clarify what traditional procedure mean in this context to be classified as recommended.

Results

14.   At the time of the survey, 42% of the women were using contraception, and 8% were using long-acting reversible contraceptives (LARCs).

·       does this 8% come from 42% of the women were using contraception. if yes, consider to reformulate the sentence

15.   The designation of the different columns presented in the table are difficult to understand.

·       Explain the meaning of 0 confident; and >= confident.

16.   Overall, about 64% of the respondents reported having a confidant living in the DRC.

·       where can we visualize the percentage indicated in this statement?

17.   Having at least one child, educational attainment, current contraceptive use, and long-acting contraceptive use were similar between the respondents and confidants.

·       table 1 appear to say something different. can you revisit it

18.   Table 3. Distribution of sociodemographic and economic characteristics of respondents aged 15 to 29 who reported an abortion and shared it with a confidant, according to type of abortion in the city of Kinshasa, DRC *.

this table is not clear for me in terms of what is represented.  the designation must be reflected in table. does this distributing socio-economic characteristic of respondents who reported an abortion and shared it with a confidant? the table also does not show results according to the type of cities. What are those cities you a referring to? I suggest the author to split the table in:

·       fist table that presents sociodemographic characteristics and type of abortion

·       second type of abortion vs shared with friend

All this presented in such way that allow us to see differences according to type of city, as you are indicating

19.   Injections (23%) and other nonmedical abortion pills (15%) were also commonly used. The most common source of care was private facilities (45%), followed by pharmacies (36%).

The use of similar designation (table vs description) helps to understand what is described. Use similar designation in both

Where is the information pharmacies (36%) in the table?

Discussion

20.   This study aimed to understand the experience of induced abortion among women residing in the slums of Kinshasa, where there is restricted availability of formal healthcare services.

·       This objective is different from this objective the author mentioned ion the abstract (The evolution of the legal and practical landscape of abortion in the Democratic Republic of the Congo (DRC) over the last few years necessitates a re-examination of the experience of induced abortion, leading this study to measure the incidence of abortion among young women (15 to 29 years of age), as well as the heterogeneity of this problem according to the residence of these young women (slum vs. non-slum areas).

if the objective of this study is to investigate how those at slums are different from the non-slums, the analysis must focus on that. what it is understood in this study is:  the variable slum is used in similar way as other social, demographic and economic characteristics to describe the experience on abortion. the analysis presented in this study make evident respondent and confident only. not respondent at slum/no-slum and confident in slum/non-slum.

·       So, it is necessary to present the main result in a way that it is evident that you are exploring differences between those living in slums and non-slums

21.   The young women included in this investigation were mostly unmarried (more than three-quarters of our group). Among the confidants, unmarried women had a higher rate of resorting to abortions than married women (156 vs. 60).

·       how to see that these unmarried women are from slum if the analysis was not conducted comparing this group of women in slum an out of slums?

22.   This is in contrast to the findings of Aké-Tano, P. et al.’s study conducted in Yamoussoukro in 2017, where the predominant strategy utilized by participants was the consumption of medical items, accounting for 92% of cases.

·       there is a need to understand what is the meaning of consumption of medical items? what does this involve?

Author Response

Authors response to Reviewer 1

Report of manuscript review

This study on Abortion Incidence among Young Women in Urban Slums and Non-Slums in Kinshasa, DR Congo is crucial to understand how women facing unintended pregnancy manage their situation. this in turn, helps to inform the decision make to take action in order to safe women life by providing safe induce abortion services. I would like to congratulate the authors for their effort. However, there are important aspects not considered in this study that could provide a more comprehensive understanding. While this study aimed to understand the experience of induced abortion among women residing in the slums of Kinshasa, where there is restricted availability of formal healthcare services, the author did not show, in the result, the experience of the participants based on categories slum and non-slum. In the table presented they explore types of induced abortion and show for whom they reported. What the importance of exploring to whom they reported? What benefit does this offer in terms of strategy of health improvement. These questions are important to be explained to make the reader in context. Bellow I present specific comments.

Abstract:

  1. normally, groups of age being used in sexual and reproductive health are 15-49 as women of reproductive age in general.When the study involves only young people the groups used are 15-19 for adolescents and 20-24 youth.
  • can the Autor explain the groups of age used in this study? What is the definition applied, from whom?

Authors response: Some of (around one in third) pregnancies among non-married women are ended by abortion. In Kinshasa, the median age at first union rose from 24.3 years to 29.5 years. While the age group of 15 to 29 years old are also sexually active, suggesting an increased risk of abortion in this age group. The idea is the estimated the incidence during this age group because at least 50% are unmarried.

  1. The incidence rate of induced abortion among the confidants (close friends) in 2021 was 131.5 per 1000 (95% CI: IQR 99.4–163.6).These rates were significantly higher than the corresponding estimates of those surveyed.
  • Which estimates, which surveyed, and by whom?

Authors response: We have re-written as follow: “The fully adjusted one-year friend abortion rate in 2021 was 131.5 per 1000 (95% CI: IQR 99.4–163.6). These rates were significantly higher than the corresponding estimates of respondents. The incidence of induced abortion for respondents was 24.4 per 1000 (95% CI: 15.8–32.9) abortions per 1000 women”

  1. There is still a lot of work to be done to fulfill the obligations outlined in the Maputo Protocol. This includes establishing easily accessible services that offer complete abortion care with a focus on women's needs, particularly in slum areas.
  • from the result described before, it is not evident this statement the Autor made

Authors response: We have edited: Slum respondent indicated higher use of non-recommended methods than non-slum respondent

Background

  1. An analysis of abortion rates and safety trends is necessary to track advancements and provide information for ongoing reforms due to the changing legal and practical situation surrounding abortion in the Democratic Republic of the Congo (DRC) in recent years.
  • can the author explain a bit more this story about abortion law before the changes and what changes are occurring now?

The availability of safe abortion in the DRC has been limited in the past due to various factors. These include conflicting laws that have caused uncertainty about its legal status, a shortage of trained medical professionals and equipment to perform the procedure, inadequate awareness among potential clients about where to access abortion services, and societal stigma surrounding the procedure. The ratification of the Maputo Protocol in 2018 and its publishing in the National Gazette should have resulted in the decriminalization of abortion and the definitive resolution of its legal status. According to Article 14 (2) (c) of the Maputo Protocol, States Parties are required to implement necessary measures to safeguard women's reproductive rights. This includes allowing medical abortion in situations involving sexual assault, rape, incest, and when the mother's mental and physical health, as well as the life of the mother or the foetus, are at risk due to the continuation of the pregnancy

  1. A significant majority of the impoverished urban population in numerous sub-Saharan nations reside in slums, which are frequently isolated from official public services, including FP programs.
  • slums correspond to peripheral areas of the cities characterized by high population density which result in high demand in services which most the time do not meet integrally the need of that population. when the author says isolated from official services, is this similar to say that they walk long distances, for example 10 km, to find health facility? Is lack of health services infra-structure or the problem is the response to demand?

 Authors response: It is mostly the lack of health services infrastructure

  1. Akilimali published abortion incidence in Kinshasa [9], These alarming statistics have prompted unprecedented policy action to reduce the morbidity and mortality associated with unsafe abortion in the DRC.Where can we see these alarming statistics in this manuscript? Can you include them to make more evident

  Authors response: This was included in the current version

  1. The aims of this research were to the heterogeneityof abortion incidence between slum and non-slum residence specifically among youth aged under 30 years.

 Seems like something is missing in this passage

 Authors response: Yes, you are right, there was a missing word; We have edited actually can be seen as follow “The aims of this research were to examine the heterogeneity of abortion incidence between slum and non-slum residence specifically among youth aged under 30 years.”

which concept of youth this study is using in this study, WHO or from other organization and why? see comment in abstract section

  Authors response: We have replaced youth by Women under 30 years

  1. These behaviors can result in unplanned pregnancy and/or the transmission of sexually transmitted illnesses.There is repetition of information

Authors response: Yes, we have edited the sentence

Methodology

  1. Data from the Performance Monitoring for Action (PMA) study were used
  • can the author explain more about this PMA

Authors response: PMA conducts nationally or regionally representative surveys on reproductive health in nine countries in Africa and Asia. PMA DRC conducts surveys in two provinces, Kinshasa and Kongo Central. It used a two-stage cluster sampling design in each province. Primary data collection selected clusters using probability proportional to size sampling, listed all households within each cluster, and randomly selected 35 households per cluster to identify a representative sample of households. More details on PMA selection procedures can be found at www.pmadata.org/data/survey-methodology. 

  1. Out of the initial database of 2326 women in Kinshasa, we specifically chose 1397 women who were between the ages of 15 and 29.
  • how these specific women were selected for this study? in the recommended place to see methodology selection process the explanation presented is about  selection process of all sample.

Authors response: All women aged 15-29 among 2326 women interviewed in Kinshasa were kept in the analysis of current manuscript.

  1. This measure was derived from the latest indicator of abortion safety provided by the World Health Organization (WHO).
  • Indicate the year

Authors response: Added: in 2022

  1. However, the least safe abortions are carried out by untrained clinicians using non-recommended methods.
  • What do we call clinicians

Authors response: We replace by provider

  1. Any techniques other than pills, injections, or traditional procedures were categorized as not recommended.
  • explain a bit more this statement to clarify what traditional procedure mean in this context to be classified as recommended.

Authors response: We classified surgical abortion and medical abortion pills (misoprostol and related) as recommended methods and all other methods (other or unknown pills, injections, traditional methods) as non-recommended. Here traditional methods are lives and other traditional practices

Results

  1. At the time of the survey, 42% of the women were using contraception, and 8% were using long-acting reversible contraceptives (LARCs).
  • does this 8% come from 42% of the women were using contraception. if yes, consider to reformulate the sentence

Authors response: We have edited as follow: “At the time of the survey, 42% of the women were using contraception (all methods), and 8% were using long-acting reversible contraceptives (LARCs).”

  1. The designation of the different columns presented in the table are difficult to understand.
  • Explain the meaning of 0 confident; and >= confident.

Authors response: We are comparing respondents with 0 confident to those with 1 + confidents

  1. Overall, about 64% of the respondents reported having a confidant living in the DRC.
  • Where can we visualize the percentage indicated in this statement?

Authors response: about 64% (890/1397)

  1. Having at least one child, educational attainment, current contraceptive use, and long-acting contraceptive use were similar between the respondents and confidants.
  • table 1 appear to say something different. can you revisit it

Authors response: Having at least one child, educational attainment, current contraceptive use, and long-acting contraceptive use were similar between the respondents with 0 confident and those with 1 + confidents (Table 2).

  1. Table 3. Distribution of sociodemographic and economic characteristics of respondents aged 15 to 29 who reported an abortion and shared it with a confidant, according to type of abortion in the city of Kinshasa, DRC *.

this table is not clear for me in terms of what is represented.  the designation must be reflected in table. does this distributing socio-economic characteristic of respondents who reported an abortion and shared it with a confidant? the table also does not show results according to the type of cities. What are those cities you a referring to? I suggest the author to split the table in:

  • fist table that presents sociodemographic characteristics and type of abortion
  • second type of abortion vs shared with friend

All this presented in such way that allow us to see differences according to type of city, as you are indicating

Authors response: We now presented only type of abortion vs shared with friend. We have only Kinshasa

  1. Injections (23%) and other nonmedical abortion pills (15%) were also commonly used. The most common source of care was private facilities (45%), followed by pharmacies (36%).

The use of similar designation (table vs description) helps to understand what is described. Use similar designation in both

Where is the information pharmacies (36%) in the table?

 Authors response: The table was edited

Discussion

  1. This study aimed to understand the experience of induced abortion among women residing in the slums of Kinshasa, where there is restricted availability of formal healthcare services.
  • This objective is different from this objective the author mentioned ion the abstract (The evolution of the legal and practical landscape of abortion in the Democratic Republic of the Congo (DRC) over the last few years necessitates a re-examination of the experience of induced abortion, leading this study to measure the incidence of abortion among young women (15 to 29 years of age), as well as the heterogeneity of this problem according to the residence of these young women (slum vs. non-slum areas). 

if the objective of this study is to investigate how those at slums are different from the non-slums, the analysis must focus on that. what it is understood in this study is:  the variable slum is used in similar way as other social, demographic and economic characteristics to describe the experience on abortion. the analysis presented in this study make evident respondent and confident only. not respondent at slum/no-slum and confident in slum/non-slum.

  • So, it is necessary to present the main result in a way that it is evident that you are exploring differences between those living in slums and non-slums

  Authors response: We have highlighted the difference between slum and non-slum in the abstract, results and discussion section

  1. The young women included in this investigation were mostly unmarried (more than three-quarters of our group). Among the confidants, unmarried women had a higher rate of resorting to abortions than married women (156 vs. 60).
  • how to see that these unmarried women are from slum if the analysis was not conducted comparing this group of women in slum an out of slums?

 Authors response: We have added table comparing slum and nonslum.

  1. This is in contrast to the findings of Aké-Tano, P. et al.’s study conducted in Yamoussoukro in 2017, where the predominant strategy utilized by participants was the consumption of medical items, accounting for 92% of case
  • there is a need to understand what is the meaning of consumption of medical items? what does this involve?

Authors response:  We have edited the statement: “where the predominant strategy utilized by participants was the consumption of medical abortion pills”

Reviewer 2 Report

Comments and Suggestions for Authors

Paper: Abortion Incidence among Young Women in Urban Slums and Non-Slums in Kinshasa, DR Congo

Induced abortion is a public health issue around the world e particular in sub-Saharan Africa. This is mainly because a significant number of abortions are unsafe which put the file of the involved women in danger. In addition, population-based abortion estimates are limited. This paper was aimed at estimating abortion rate in slams and non-slams urban areas of the city of Kinshasa (DR Congo).

The paper uses survey data of PMA study conducted between 2021 and 2022 where the surveyed respondents were asked about their own experience and that of their confidents (close friends). However, the reason for asking the respondents about the experience of their close friends or the possible bias resulting from the fact that one knows its own experience better than that of somebody else even if that person is a close friend is not discussed. I think this is important because a comparison between abortion rates from respond experience and that of their confidants is made and some of the difference may account for that bias.

The results are clearly presented and are consistent with what would be expected with regard to differences between adolescent and young women in slams and non-slams areas. However, it is intriguing why rates for the confidents are much higher than those of the respondents themselves.

The discussion is well written and show that the results are consistent with finding elsewhere, but little is included in the discussion regarding the confidents’ results. It would be important to include some information in the discussion about this particularly because the rates are estimated based on someone else’s report and some bias may be associate with it. They are also consistent with other

Author Response

Paper: Abortion Incidence among Young Women in Urban Slums and Non-Slums in Kinshasa, DR Congo

 Induced abortion is a public health issue around the world e particular in sub-Saharan Africa. This is mainly because a significant number of abortions are unsafe which put the file of the involved women in danger. In addition, population-based abortion estimates are limited. This paper was aimed at estimating abortion rate in slams and non-slams urban areas of the city of Kinshasa (DR Congo). 

The paper uses survey data of PMA study conducted between 2021 and 2022 where the surveyed respondents were asked about their own experience and that of their confidents (close friends). However, the reason for asking the respondents about the experience of their close friends or the possible bias resulting from the fact that one knows its own experience better than that of somebody else even if that person is a close friend is not discussed. I think this is important because a comparison between abortion rates from respond experience and that of their confidants is made and some of the difference may account for that bias.

 The results are clearly presented and are consistent with what would be expected with regard to differences between adolescent and young women in slams and non-slams areas. However, it is intriguing why rates for the confidents are much higher than those of the respondents themselves.

Authors response: Due to social desirability bias, estimating abortion incidence among respondent would underestimate the abortion incidence. Using friends or confident gives us the opportunity to talk about the abortions of relatives and enables us to estimate the values closest to reality

The discussion is well written and show that the results are consistent with finding elsewhere, but little is included in the discussion regarding the confidents’ results. It would be important to include some information in the discussion about this particularly because the rates are estimated based on someone else’s report and some bias may be associate with it. They are also consistent with other

Authors response:

Although we may have indirectly measured abortion through closest, we acknowledge the presence of bias. Nevertheless, the estimated value obtained through closest is more accurate than the estimate provided by the respondents.

Reviewer 3 Report

Comments and Suggestions for Authors

Overall this is a very well written paper. Th findings are important and will contribute to public health policy in sub Saharan Africa.

The paper examined the experience of induced abortion and measured the incidence of abortion among young women (15 to 29 years of age), as well as the heterogeneity of this problem according to the residence of these young women (slum vs. non-slum areas).

The study is very significant as it enables readers to understand to establish the incidence of abortion among young women (15 to 29 years of age) and their confidants. This is really crucial for public policy on reproductive education in sub Saharan Africa.

The paper made a significant contribution to our understanding of induced abortion among young women (15 to 29 years of age) and their confidants. There are not many studies that focus on these two demographics.

This paper added more granularity to the existing data. The paper drilled down to slums and non-slums in Kinshasa. Other papers would mostly give a generical statistical overview. The evidence provided had depth.

The methods was ok to me.

Conclusion is consistent with the evidence although it was very brief. The authors captured most of the conclusions in the discussion section. So I would recommend that the author (s) add to the conclusion section.

The references are appropriate.

The paper will require minor edits as I picked up a few typographical errors.

Comments on the Quality of English Language

The paper will be ok to publish subject to minor edits and proof reading

Author Response

Overall this is a very well written paper. Th findings are important and will contribute to public health policy in sub Saharan Africa.

The paper examined the experience of induced abortion and measured the incidence of abortion among young women (15 to 29 years of age), as well as the heterogeneity of this problem according to the residence of these young women (slum vs. non-slum areas).

The study is very significant as it enables readers to understand to establish the incidence of abortion among young women (15 to 29 years of age) and their confidants. This is really crucial for public policy on reproductive education in sub Saharan Africa.

The paper made a significant contribution to our understanding of induced abortion among young women (15 to 29 years of age) and their confidants. There are not many studies that focus on these two demographics.

This paper added more granularity to the existing data. The paper drilled down to slums and non-slums in Kinshasa. Other papers would mostly give a generical statistical overview. The evidence provided had depth.

The methods was ok to me.

Conclusion is consistent with the evidence although it was very brief. The authors captured most of the conclusions in the discussion section. So I would recommend that the author (s) add to the conclusion section.

The references are appropriate.

The paper will require minor edits as I picked up a few typographical errors.

Comments on the Quality of English Language

The paper will be ok to publish subject to minor edits and proof reading

Authors response: Thanks for the kind words. We have added this statement in conclusion section: The incidence of induced abortion for respondents is higher in slum compared to non-slum. Slum respondents have higher use of non-recommended methods than non-slum respondent.
